# Wine with Added Pomegranate Juice: A Novel Approach to Sparkling Winemaking

**DOI:** 10.3390/foods14040581

**Published:** 2025-02-10

**Authors:** Shuyan Liu, Simone Vincenzi

**Affiliations:** Interdepartmental Centre for Research in Viticulture and Enology, University of Padova, 31015 Conegliano, Italy; shuyan.liu@unipd.it

**Keywords:** sparkling wine, pomegranate juice, polyphenols, antioxidant activity, Martinotti method, wine innovation

## Abstract

This study investigates the incorporation of pomegranate juice into the fermentation process to produce a novel rosé sparkling wine with enhanced antioxidant properties and improved acidity while preserving desirable sensory qualities. Initial trials blended a Glera base wine with 5%, 10%, and 15% pomegranate juice, followed by bottle fermentation to select the optimal formulation. The 10% blend, identified as the best, underwent a second fermentation in an autoclave using the Martinotti method. Chemical analyses were conducted to assess polyphenol content, protein stability, antioxidant activity, and colour, while sensory evaluations determined the flavour balance and acceptability. Results demonstrated that pomegranate juice significantly increased polyphenol content and antioxidant capacity. The 10% blend achieved balanced acidity, sweetness, and carbonation, with pronounced fruity and floral aromas. However, protein stability tests revealed haze formation, emphasizing the need for stabilization refinements. This study highlights the feasibility of integrating pomegranate juice into sparkling wine production. It explores alternative and innovative uses for pomegranate to maximize its potential beyond traditional applications, leverages its unique properties, such as high antioxidant content, to enhance value-added products, and demonstrates its potential to diversify the portfolio of fruit wines while appealing to modern consumer preferences.

## 1. Introduction

The pomegranate (*Punica granatum*) fruit has a round shape and is composed of the exocarp (peel), mesocarp, endocarp, and grains. The grains, which form the edible part of the fruit (approximately 60% of the fruit’s total weight), consist of a seed covered by a casing of fleshy, juicy, and reddish pulp called the aril. The arils contain high levels of polyphenols, minerals, vitamins, and organic acids, especially malic and citric acids associated with the sensory perception of acidity, as well as two types of sugars, fructose and glucose [1].

Currently, there is growing interest in the pomegranate fruit due to its high antioxidant content, particularly polyphenols such as punicalagins, anthocyanins, and flavonoids. These compounds have long been associated with various medicinal properties [2,3] and have earned pomegranate recognition as a functional food due to its health benefits [4,5]. However, the perceived difficulty of extracting the arils from the fruit still limits the consumption of fresh pomegranate [6], encouraging producers to focus on pomegranate juice production. Even in this case, the diffusion of large amounts of hydrolysable tannins in commercial pomegranate juices may pose challenges for sensory quality, as they induce a strong, unpleasant, and astringent mouthfeel [7].

The production of pomegranate wine has emerged as a novel approach to utilize this fruit [1]. However, despite the minimal loss of polyphenol content during processing [8,9], the disappearance of sugar through fermentation intensifies the perception of sour, bitter, and astringent components [10].

Sparkling wines have become increasingly important in the global wine market due to their distinctive sensory qualities and broad appeal. Additionally, non-grape fruit sparkling wines, particularly those made from plum and apple [11], as well as sparkling wines with added fruit juice [12], have gained attention. These wines are produced through a secondary fermentation process that creates their characteristic bubbles. The traditional “Champagne method” is the most recognized technique for producing high-quality sparkling wines [13]. However, the Martinotti method, where fermentation occurs in a closed tank, has gained popularity, particularly with the rise of Prosecco wine. A key difference between the two methods is the shorter contact with yeast lees in the Martinotti process, resulting in wines that are fresher and more expressive of their varietal characteristics [14]. In parallel, rosé wines have gained popularity due to their versatility. Rosé wines are typically produced by limiting the contact between the grape skins and juice, yielding a lighter colour and a more delicate flavour profile compared to red wines. This process involves either short maceration or direct pressing techniques [15].

The rapidly growing market for health-focused drinks, driven by increasing consumer demand for antioxidant-rich options, underscores the need for innovative products [16]. This study investigates the potential of using pomegranate juice in the autoclave refermentation of a base white wine to create a rosé sparkling wine that combines enhanced antioxidant properties with improved acidity and flavour. Utilizing pomegranate in this way highlights its unique properties and adds value through innovative applications. Integrating pomegranate juice into sparkling wine production is a significant step toward diversifying the portfolio of fruit wines while meeting modern consumer preferences for healthier and more innovative beverages.

## 2. Materials and Methods

### 2.1. Pomegranate Juice and Base Wine

The pomegranate juice was provided by Azienda Agricola Maira Marcello (Caltanissetta, Italy) and was obtained by pressing ripe fruits of the Dolce di Sicilia variety. Microbiological stabilization of the juice was achieved through pasteurization. The base wine, a single-variety wine from Glera, was provided by Instituto Cerletti, Conegliano (Italy).

### 2.2. Sparkling Wine Production

The yeast strain Safoeno BC S103 (Fermentis, Marquette-lez-Lille, France) was used to produce sparkling wine. It was prepared as follows: the dried yeast (20 g/hL calculated in the final volume of wine) was rehydrated in 10 volumes of water at 37 °C containing an equal weight of Go Ferm Protect (Lallemand, Montreal, Canada). After about 30 min, the yeast was cooled to 25 °C, and 4 volumes of base wine and 4 volumes of water were added. The conventional oenological parameters of the base wine of the Glera variety are shown in Table 1. This mixture contained a final concentration of 1.5 g/L of diammonium phosphate, 0.01 mg/L of thiamine, and 80 g/L of sugars. After 12 h, the yeast was further cooled to 20 °C and 2.5 volumes of base wine and 1 volume of water were added. This addition contained a final concentration of 0.3 g/L of diammonium phosphate, 0.01 mg/L of thiamine and 50 g/L of sugars. After a further 24 h, the inoculum was added to the base wine, which was obtained from Glera variety. The inoculum (representing approximately 8% of the total mass) had been previously mixed with different percentages of pomegranate juice.

In the first step, a small-scale trial for sparkling wine production was conducted through bottle fermentation. A Glera base wine was mixed with different volumes (5, 10, and 15%) of pomegranate juice and inoculated with the prepared yeast. To compensate for the different sugar content of the three base wines (resulting from different pomegranate juice percentages) and ensure similar final alcohol content and pressure development, different quantities of rectified and concentrated grape juice were added. After the addition of yeast inoculum to each mixture, bottling was carried out in 375 mL bottles, sealed with a crown cap and stored vertically in a cellar at 18 °C to initiate refermentation. Three bottles from each trial were opened every two days and analyzed for sugar to monitor the fermentation rate.

In the second step, the refermentation of a blend consisting of the base wine with 10% pomegranate juice was repeated in a small autoclave (30 L, Cadalpe, Vazzola, Italy). This setup allowed for filtration and residual sugar retention, producing a product closer to the finished version. The autoclave was maintained at 18 °C, and samples were collected every two days to analyze sugar levels.

At the end of fermentation, the wine was filtered using polyether sulfone cartridges (0.5 µm, Pall Corporation, Port Washington, NY, USA), adjusted with potassium metabisulfite to achieve at least 25 mg/L of free sulphur dioxide and brought to 12 g/L of sugar (extra dry version). The wine was then bottled in 0.75 L white bottles. The bottles were stored at 4 °C in the dark until sensorial analysis.

### 2.3. Conventional Oenological Parameter

The alcohol content was determined using a Gibertini (Novate Milanese, Italy) Oenochemical distilling unit (Italy). pH was measured with a pH meter (model Accumet AB315, Fisher Scientific, Segrate, Italia). Total acidity was measured by the titrimetric method with NaOH and bromothymol blue (OIV-MA-AS313-01) [17]. Free and total sulphur dioxide were determined by iodine titration in presence of starch indicator (OIV-MA-AS323-04B) [17]. Acetic acid, sugar, and acetaldehyde concentrations were quantified using an enzymatic auto analyzer (iMagic-M9, R-Biopharm, Milano, Italy).

### 2.4. Colour Analysis

Wine colour was measured at 420 nm (yellow), 520 nm (red), and 620 nm (blue) using an Ultrospec 2100 pro UV-Vis spectrophotometer (Euroclone, Milano, Italy) after filtering at 0.45 µm. The optical densities were measured against water in quartz cuvettes with 1 mm or 10 mm optical length.

### 2.5. Polyphenol Analysis

Total polyphenols were determined by the Folin–Ciocalteu method [18] in mg/L of gallic acid. Absorbance at 280 nm provided additional information on phenolic compounds based on their UV absorbance, complementing the total polyphenol measurement. Absorbance at 320 nm measured hydroxycinnamic acids and flavonols, offering more specific insights into compounds related to antioxidant capacity and oxidative stability.

### 2.6. Protein Precipitation and Quantification

Pomegranate juice was filtered at 0.45 µm, and soluble proteins were precipitated and quantified following the KDS method [19]. Protein isolation from 1 mL of sample was achieved by precipitating with 10 µL of 10% Sodium Dodecyl Sulphate (SDS) followed by 250 µL of 1 M KCl. The protein pellet was resuspended in 1 mL of distilled water and quantified using the BCA-200 (Pierce, Rockford, IL, USA) protein assay with a bovine serum albumin (BSA, Sigma, Milan, Italy) standard curve.

### 2.7. Protein Stability Test

The method proposed by Pocock e Rankine (1973) [20] was applied. All the samples were filtered, and the turbidity was measured using a nephelometer (Hanna Instruments, Italy). Then, 15 mL of each sample was heated in an oven at 80 °C for 6 h, followed by refrigeration at 4 °C for 12 h. Turbidity was measured again, and the sample was considered stable if the difference between the measurements before and after heating was less than 2 NTU (Nephelometric Turbidity Unit).

### 2.8. Antioxidant Activity

The antioxidant activity of the samples was determined by the DPPH (1,1-diphenyl-2-picrylhydrazyl) discoloration test [21]. DPPH was prepared dissolving 2.5 mg in 100 mL of methanol. For each sample, three replicates were prepared by adding 3 μL of pomegranate juice mixed with 7 μL of water, or 10 μL of wine sample, to 2 mL of DPPH solution in a plastic cuvette, ensuring consistent sample volume and preventing the antioxidant activity of pomegranate juice from exceeding the upper limit of measurement. The decrease in absorbance at 517 nm was measured after exactly 10 min of incubation at room temperature. A blank was prepared by replacing the sample with water, and quantification was performed by comparison with a calibration curve obtained by adding 1 to 15 μL of Trolox to the DPPH solution. The results were expressed as Trolox equivalents (TEAC).

### 2.9. Sensory Analysis

To evaluate the organoleptic characteristics of the wines blended with different percentages of pomegranate juice and fermented in bottles, a sensory evaluation was conducted with five panellists experienced in winemaking and tasting using the flash profiling method [22]. Based on the results, an optimal percentage of pomegranate juice was selected for high-volume autoclave fermentation. To determine the sensory profile of the final autoclave-fermented wine, a trained panel of twelve assessors (four females and eight males), aged 29 to 63, performed a descriptive evaluation of the wine samples. All panellists had extensive experience in the sensory evaluation of sparkling wines. The assessors were asked to rate the intensity of the wine’s colour (purplish-red), aroma (floral, fruity, beet, red berry, cherry, and pomegranate), taste (bitter, sour, and sweet), and mouthfeel (astringency). The list of sensory attributes was generated based on previous research on pomegranate juice and wines [23]. Approximately 30 mL of wine was served to each panellist. Unsalted crackers and water were provided for palate cleansing between samples. The evaluation was conducted at the CIRVE (Interdepartmental Centre for Research in Enology and Viticulture) of the University of Padova, located at the Conegliano campus (Italy). The facilities were equipped with individual booths under controlled lighting and temperature conditions.

### 2.10. Statistical Analysis

To determine the statistical significance of differences between the results, data were analyzed using analysis of variance (ANOVA), followed by Tukey’s HSD test for multiple comparisons. Statistical significance was set at *p* < 0.05. All analyses were performed using JMP Pro 17 software (SAS Institute, Cary, NC, USA).

## 3. Results and Discussion

### 3.1. Pomegranate Juice and Base Wine Analysis

The results of the analysis of pomegranate juice and Glera base wine are reported in Table 1. The titratable acidity was expressed in citric acid, as it is the acid most commonly found in pomegranate juice [24,25]. The value is notably high compared to those reported in the literature, which range from 12.5 g/L [25], 18 g/L [26] and 16.3–25.6 g/L [27] to 18–28 g/L [28]. This observation aligns with the pH, which was lower than the values found in the literature, reported as 3.13–3.30 [27], 3.1 [26], and 3.94–4.07 [28]. These results cannot be attributed to poor fruit maturity, as the sugar levels are consistent with those reported in other studies: 133–177 g/L [27] and 151 g/L [25]. In fact, the sugar content is even higher than values reported by Legua and collaborators in 2012 (137–148 g/L) [28]. Therefore, these results likely reflect the characteristics of the fruits used to produce the juice. The high acidity of pomegranate juice is a beneficial attribute for its application in sparkling wine production, as acidity is a critical factor for maintaining the freshness of the final product.

The phenolic content of pomegranate juice was exceptionally high. The polyphenol content exceeds values typically reported in the literature (784.4–1551.5 mg GAE/L or 604-1208 mg/L quercetin equivalent). However, a recent study conducted on the Wonderful variety grown in Sicily revealed a similar polyphenol content (2476 mg/L GAE) [27], comparable to the sample analyzed in this paper. Such differences with previous data from the literature may depend on varietal differences, which significantly affect the phenolic content of juice [1,29].

### 3.2. Sparkling Wine Production in Bottle

A preliminary trial of sparkling pomegranate wine production was conducted using bottle fermentation to determine the optimal ratio between pomegranate juice and the base wine from a sensory perspective. Three different percentages of pomegranate juice were added to the base white wine. Prior to yeast addition, an aliquot of each mixture was analyzed to measure acidity and pH (Table 2), which, unlike sugars, were not adjusted. As expected, the addition of increasing quantities of pomegranate juice resulted in a decrease in pH and an increase in titratable acidity.

After yeast inoculation, the fermentation rate was monitored by opening three bottles from each trial every two days to measure sugar levels. As shown in Figure 1, fermentation concluded after 14 days. The refermentation rate in the three tests was very similar and consistent. However, the sample with the highest juice content (15%) appeared to ferment more efficiently and rapidly. Since no additional nitrogen was supplied during yeast inoculation, it can be hypothesized that a higher juice concentration provided more nutritional compounds (e.g., nitrogen and vitamins), enhancing yeast efficiency in converting sugars to alcohol. In any case, all wines completed refermentation, achieving residual sugar levels below 2 g/L.

Alcohol content and total acidity were measured at the end of fermentation. As shown in Table 2, the final alcohol content was similar among the different samples, confirming that the sugar adjustment performed at the time of yeast inoculation was accurate. Regarding total acidity, acidity decreased slightly after refermentation, which was expected due to the increase in alcohol causing a reduction in potassium bitartrate solubility. In the 15% sample, this reduction was not observed—perhaps because dilution with pomegranate juice (containing citric and malic acids but not tartaric acid) lowered the concentration of tartaric acid, reducing the solubility product below saturation. This observation could be an interesting factor to consider, as using pomegranate juice above certain percentages may naturally stabilize the wine, potentially reducing the need for tartaric stabilization after sparkling wine production.

### 3.3. Protein Stability and Quatification

The base wine, pure pomegranate juice, and the three finished wines were tested for protein stability. As shown in Table 3, while the base wine is definitively stable (the delta NTU before and after heating is less than 2), the wines with added pomegranate juice developed a high level of turbidity. Between 5% and 10%, there was still an increase in turbidity, which, however, was not observed with a further increase from 10% to 15% pomegranate juice. Pure pomegranate juice demonstrated only a slight increase in turbidity, in agreement with the data of Cerreti and colleagues (2016) [30], who reported, using the same heating protocol, an increase in turbidity of approximately 40 NTU. However, this does not completely exclude the presence of proteins in pomegranate juice, as heat precipitation also depends on various other factors [31]. The limited turbidity development cannot be attributed exclusively to a lower protein content. Moreover, the values measured in wines with added pomegranate juice were significantly high, making it even less likely to exclude the presence of a high protein content in pomegranate juice.

In the literature, data on the protein content of pomegranate juice vary widely depending on the cultivar, extraction method, and, most importantly, the analysis technique used. Protein analysis of pomegranate juice is often performed with the Kjeldahl method, which measures total nitrogen, including that present in non-protein forms (e.g., ammonium, amino acids, peptides). This results in an overestimation of actual protein content. Rinaldi et al. (2013) [32] found values between 2400 and 4900 mg/L using the Kjeldahl method. A widely used method for protein analysis in plant matrices is the Bradford assay [33], which uses Coomassie Blue dye and can measure proteins even in the presence of high polyphenol concentrations. Using this method, Tastan and Baysal (2015) [34] reported significantly lower values, between 355 and 377 mg/L in fresh juice, which decreased to 37.4-61.9 mg/L after 6 months due to precipitation with tannins and degradation. However, it should be noted that these values may be underestimated, as the Bradford method still experiences interference from polyphenols, as demonstrated in wine studies [35]. Di Nunzio and colleagues (2013) [36] used the Lowry method, which relies on the reducing capacity of proteins toward copper. This method is highly susceptible to interference from high polyphenol contents, leading to inflated values (e.g., 13,800 mg/L).

In this study, a method developed for wine, called KDS (Vincenzi et al., 2015) [35], was used. It has been proven to be more accurate and less susceptible to interference than the Bradford method. The KDS method has already been applied to fresh pomegranate juice [30], yielding values of 690–720 mg/L of protein. Using the KDS method, the pomegranate juice in this study was measured to have a protein content of 139 ± 43.7 mg/L. This lower protein content compared to values found in the literature could be attributed to pasteurization, which may have caused the denaturation and precipitation of some soluble proteins.

Assuming a protein content of 139 mg/L, adding juice at 5%, 10%, and 15% corresponds to final concentrations of approximately 7, 14, and 21 mg/L of proteins in the product, respectively. Based on data from the literature, a wine is considered unstable when it has more than 20 mg/L of protein. Therefore, the addition of pomegranate juice should not have caused such a considerable increase in turbidity. The most probable hypothesis is that the phenolic compounds in pomegranate juice (particularly tannins) interacted with the macromolecular components in the wine, causing aggregation and precipitation. This is an important factor to consider for the future production of this type of sparkling product, where the base wine needs to be correctly fined to prevent haze formation during storage.

### 3.4. Colour Analysis and Sensory Evaluation of Bottle-Fermented Wines

The results of the colour analysis are shown in Figure 2. As the percentage of added pomegranate juice increases in bottle-fermented wines (5%, 10%, and 15%), the colour intensity also increases, reflected by higher absorbance values at 420 nm, 520 nm, and 620 nm. The most notable changes are observed at 420 nm and 520 nm. The addition of pomegranate juice resulted in a proportional increase in colour intensity.

The anthocyanins present in pomegranate juice differ from those found in wine. In pomegranate, delphinidin, cyanidin, and, in smaller quantities, pelargonidin are prevalent, mostly in the 3,5-diglucoside form and, to a lesser extent, in the 3-glucoside form [27,37]. Notably, pomegranate anthocyanins do not include acylated forms. In contrast, wine anthocyanins primarily consist of delphinidin, cyanidin, peonidin, petunidin, and malvidin, mainly in the 3-glucoside form. The proportions vary depending on the grape variety, with malvidin-3-glucoside generally predominating. Small and variable percentages of acylated and coumarylated anthocyanins are also present in wine.

This difference in composition could influence the stability of the colour over time, which is a critical factor in a “rosé” product. Acylation is known to enhance anthocyanin stability by protecting them from hydration [38]. This factor might suggest a greater stability of wine anthocyanins. However, it should be noted that acylated anthocyanins typically account for only a small proportion of total grape anthocyanins.

Additionally, methoxylation of ring B also improves anthocyanin stability [39]. This results in greater stability for malvidin (2 OCH3), followed by petunidin and peonidin (1 OCH3), and then by other anthocyanins. This aspect would again favour the greater stability of wine anthocyanins (with a predominance of malvidin) compared to those of pomegranate, which are mainly composed of non-methoxylated anthocyanins.

On the other hand, di-glycosylation appears to have a stronger stabilizing effect than mono-glycosylation [40]. This characteristic would favour the stability of pomegranate anthocyanins compared to those in grapes.

Unfortunately, the multitude of factors influencing colour stability, including matrix composition (pH, temperature, presence of metals, and other phenols that may participate in co-pigmentation), prevents us from determining a priori whether anthocyanins from pomegranate provide greater stability over time compared to those derived solely from grapes.

The quantities of wine obtained in the initial phase were insufficient for a comprehensive organoleptic test. However, an internal laboratory tasting revealed that in the 15% blend, the astringency was excessive, resulting in an unbalanced product. Conversely, the 10% blend produced a well-structured product that was not overly astringent, while still allowing the fruity notes of the pomegranate to be clearly perceived.

### 3.5. Sparkling Wine Production in Autoclave

Once it was established, based on bottle tests, that the best combination corresponded to a blend of the base wine with 10% pomegranate juice, this approach was repeated with a refermentation in an autoclave. This method allowed for filtering and correcting the residual sugar before bottling. With the addition of 10% pomegranate juice, the final sugar content in the mass before fermentation was sufficient to obtain 4.5 bars of pressure. Therefore, it was decided not to further increase the sugar content with exogenous sugar addition.

The analysis of sugars showed that fermentation was very slow during the first three days, but the rate increased in the following days. Fermentation was considered complete after 20 days, with final sugar levels measured at 1.08 g/L. The wine was then filtered under isobaric conditions (final cartridge 0.5 µm), brought to 0 °C, and analyzed.

As shown in Table 4, the final alcohol content of the product was quite low, only slightly below the theoretical value expected based on the quantity of sugars introduced with the pomegranate juice (10.45%). The pH and total acidity were, respectively, lowered and raised compared to the base wine, as expected, given the addition of pomegranate juice.

The relatively low acetaldehyde and acetic acidity values indicated that the yeast did not experience significant stress during fermentation. Lactic acid analysis confirmed that malolactic fermentation had not started.

A spectrophotometric analysis was also performed on the wine, comparing it with the base wine and intermediate processing phases. As shown in Table 5, the addition of pomegranate juice significantly increased all the analyzed components, particularly the absorbance at 280 nm, which corresponds to phenolic compounds in general. It is also evident that processing results in a reduction in phenolic components, particularly the colour, which is an important parameter to consider in a rosé wine.

Fermentation had the greatest impact on colour. While the reduction in absorbance at 320 nm was only 13%, the reduction in red colour (520 nm) was as much as 35%. This may be attributed to a partial degradation of anthocyanins by glycosidase enzymes, but more importantly, to the adsorption of anthocyanins onto yeast cell walls. Filtration further reduced the colour, leading to an additional 20% decrease in absorbance at 520 nm, compared to an 8.8% reduction in absorbance at 320 nm.

### 3.6. Antioxidant Activity

One of the factors that gives pomegranate juice its beneficial properties is its high content of polyphenols, which generally translates into a high antioxidant capacity. The antioxidant power value found in the pomegranate juice used in the present experiment was 3355 mg/L TEAC (Figure 3). Seeram et al. (2008) [41], comparing different drinks rich in polyphenols, found that pomegranate juice has by far the highest antioxidant power (10,412 mg/L in Trolox equivalents (TEAC)), followed by red wine (on average 4680 mg/L TEAC). Mousavinejad et al. (2009) [42] found similar values when analyzing the antioxidant power of eight different pomegranate juices, which ranged from 4655 to 10712 mg/L TEAC. Similarly, Fisher et al. (2011) [43] reported values between 5431 and 19122 mg/L TEAC. It appears that the antioxidant activity measured in the pomegranate juice used in the present paper is slightly lower than the literature data. This could be due to a loss of polyphenols caused by pasteurization, but it is still in line with the antioxidant power of red wines.

Further analysis by Pellegrini et al. (2003) [44] found antioxidant power values of 2237–3035 mg/L TEAC in red wines, 380–605 mg/L in rosé wines, and 402–485 mg/L in white wines. The TEAC value found in the Glera base wine (109.5 mg/L TEAC) is, therefore, lower than the literature values. This is justified since base wines for sparkling wines are specifically processed to have the lowest possible polyphenol content to avoid bitter or astringent sensations in the finished product. The addition of 10% pomegranate juice during refermentation, despite the losses suffered during the various phases of the process, practically quadrupled the antioxidant power of the wine, bringing it to 413 mg/L TEAC (Figure 3), a value in line with those reported for grape rosé wines.

### 3.7. Sensory Evaluation with Trained Panel

The sensory evaluation of the sparkling wines was performed by a descriptive analysis. The attributes were chosen among those suggested by Andreu-Sevilla et al., (2013) [23] for pomegranate juice and wines. The results showed that the wine was characterized by both floral and fruity notes (Figure 4). In particular, among the fruity aromas, the one associated with pomegranate was the most perceivable, together with a secondary note of red berries, demonstrating the evident contribution of the 10% pomegranate juice to the aromatic bouquet of the final product. Interestingly, the earthy note of beet, sometimes perceived in pure pomegranate juices and wines [23], was not detected in the final sparkling wine, suggesting the aroma compounds responsible for this aroma were below their detection limit due to the dilution effect. Regarding the taste, the most strongly perceived flavour was sourness, due to the increase in acidity upon pomegranate juice addition. According to the comments of the panellists, however, this acidity was not considered too high, but on the contrary, it was well balanced with the carbon dioxide of the sparkling wine and with the residual sugar of 12 g/L. Nowadays, this increase in acidity obtained with the addition of pomegranate juice can even be considered a positive aspect, as it can be exploited to counteract the decrease in grape titratable acidity experienced as an effect of global warming.

Considering the large increase in total polyphenols (increase in absorbance at 280 nm in Table 5), an increase in bitterness would also have been expected. However, the perception of bitter taste by the panellists was very low. This can be related to the relatively low content of catechins, which are the main factor responsible for bitter taste, in pomegranate juice compared to red and rosé wines [45]. Punicalagin, the main ellagitannin of pomegranate, can also contribute to the bitterness [46], but the dilution during the winemaking process probably reduces its perceptibility. On the contrary, even with a dilution factor of 10, astringency was still perceived by the panellists, with a medium-high intensity. Even though this astringency could be considered above the acceptability limit for a grape rosé wine, the panellists stated that this astringency level was not disturbing; they considered it to still be well balanced with the sugar content and to contribute to the structure of the wine.

## 4. Conclusions

This study demonstrates the successful integration of pomegranate juice into the refermentation process, offering a novel approach to producing rosé sparkling wine that meets contemporary consumer preferences for health-oriented beverages while preserving the sensory qualities expected of sparkling wines. The addition of pomegranate juice significantly enhanced the polyphenol content and antioxidant activity, delivering potential health benefits alongside a balanced flavour profile characterized by refreshing acidity and fruity aromas. The optimal formulation, with 10% pomegranate juice, achieved a harmonious blend of acidity, sweetness, and carbonation, avoiding excessive astringency or bitterness. Addressing a notable gap in the literature, this research highlights pomegranate juice as a viable functional ingredient for improving the phenolic composition and antioxidant capacity of sparkling wines. The findings expand our understanding of how fruit-based ingredients can diversify wine profiles and introduce an innovative strategy to counteract the impact of climate-related acidity reduction in grape wines. Future research should focus on optimizing stabilization techniques to enhance protein stability and investigating long-term colour retention to ensure product consistency. This study paves the way for the development of antioxidant-enhanced sparkling wines that respond to evolving consumer demands for healthier and more diverse beverage options.

## Figures and Tables

**Figure 1 foods-14-00581-f001:**
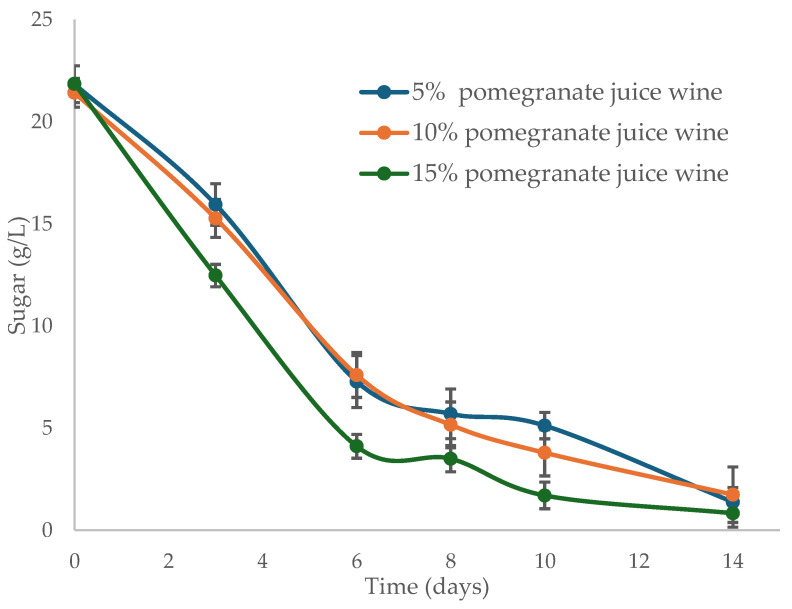
Evolution of sugar during the bottle fermentation of wines with varying pomegranate juice percentages (*n* = 2).

**Figure 2 foods-14-00581-f002:**
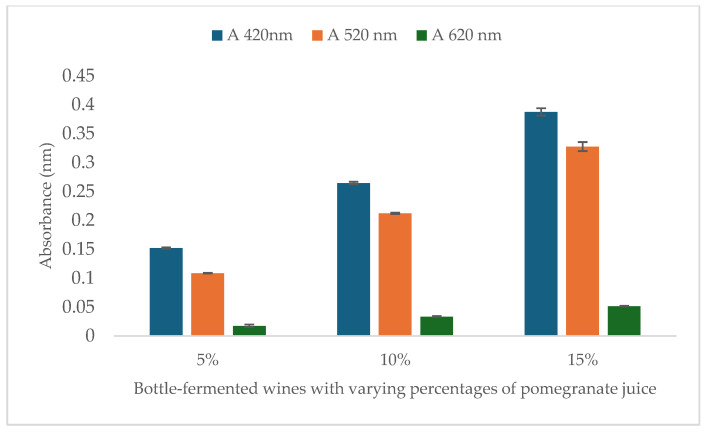
Colour results of bottle-fermented wines.

**Figure 3 foods-14-00581-f003:**
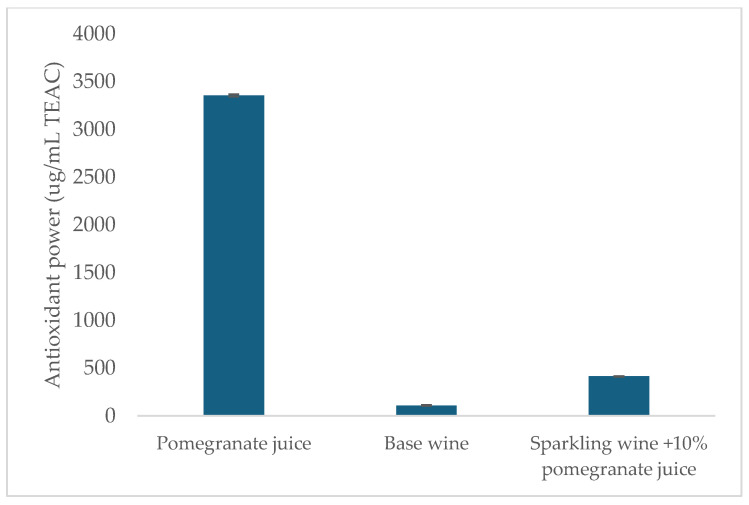
Antioxidant activity results of pomegranate juice, Glera base wine, and 10% pomegranate juice wine.

**Figure 4 foods-14-00581-f004:**
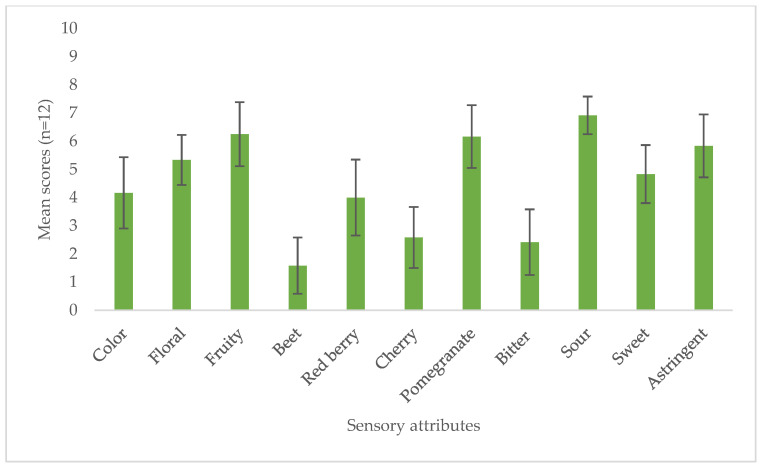
Mean sensory ratings of the 10% pomegranate juice wine.

**Table 1 foods-14-00581-t001:** Analysis of pomegranate juice and base wine.

Parameter	Pomegranate Juice	Base Wine
Alcohol (% *v*/*v*)	-	10.65 ± 0.12
pH	2.96 ± 0.02	3.41 ± 0.02
Sugar (g/L)	163.00 ± 7.15	0.25 ± 0.16
Titratable acidity (g/L)	33.50 ± 1.25 ^1^	5.80 ± 0.04 ^2^
Lactic acid (g/L)	-	0.06 ± 0.04
Acetic acidity (g/L)	-	0.12 ± 0.03
Total polyphenols (mg/L) ^3^	2788.54 ± 74.31	-

^1^ expressed in g/L of citric acid; ^2^ expressed in g/L of tartaric acid; ^3^ expressed in mg/L of gallic acid.

**Table 2 foods-14-00581-t002:** Analysis of wines with varying pomegranate juice percentages pre- and post-fermentation.

	Pre-Fermentation		Post-Fermentation	
Samples	pH	Total Acidity (g/L)	Alcohol (%*v*/*v*)	Total Acidity (g/L)
Base wine	3.41 ± 0.01 a	5.80 ± 0.04 bc	-	-
5%	3.21 ± 0.02 b	6.40 ± 0.05 b	10.55 ±0.25 a	6.00 ± 0.05 b
10%	3.19 ± 0.01 b	7.30 ± 0.09 ab	10.43 ± 0.16 a	7.00 ± 0.07 ab
15%	3.17 ± 0.01 b	8.00 ± 0.10 a	10.48 ± 0.21 a	8.00 ± 0.11 a

Different letters represent a significant difference in the Tukey HSD post hoc test (*p* < 0.05).

**Table 3 foods-14-00581-t003:** Results of protein stability.

Samples	Delta NTU
Base wine	0.47 ± 0.12 d
Pomegranate juice	16.40 ± 1.20 c
5%	452.70 ± 25.40 b
10%	603.40 ± 27.90 a
15%	560.50 ± 31.20 a

Different letters represent a significant difference at the Tukey HSD post hoc test (*p* < 0.05).

**Table 4 foods-14-00581-t004:** Analysis of 10% pomegranate juice wine after autoclave fermentation.

Parameter	10% Pomegranate Juice Wine
Alcohol (% *v*/*v*)	10.28 ± 0.13
pH	3.35 ± 0.02
Sugar (g/L)	1.08 ± 0.91
Titratable acidity (g/L) ^1^	7.30 ± 0.41
Acetaldehyde (mg/L)	10.60 ± 2.61
Lactic acid (g/L)	0.06 ± 0.06
Acetic acidity (g/L)	0.28 ± 0.09

^1^ expressed in g/L of tartaric acid.

**Table 5 foods-14-00581-t005:** Colour and polyphenol-related parameters of 10% pomegranate juice wine.

Samples	280 nm	320 nm	420 nm	520 nm	620 nm
Base wine	1.242	0.600	0.108	0.077	0.032
Base wine + 10% pomegranate juice	>3	1.115	0.399	0.340	0.082
Sparkling wine at the end of fermentation	2.602	0.970	0.259	0.187	0.046
Sparkling wine after filtration	2.321	0.884	0.205	0.139	0.032

## Data Availability

The original contributions presented in this study are included in the article. Further inquiries can be directed to the corresponding author.

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
