# Peer review of "Wine with Added Pomegranate Juice: A Novel Approach to Sparkling Winemaking"

_foods, 2025, doi:10.3390/foods14040581_

Round 1
Reviewer 1 Report
Comments and Suggestions for Authors
In this paper, the authors prepared a wine with the addition of pomegranate juice. The detailed comments are as follows:
1. Line 29, Punica granatum should be italic.
2. Add some reference in Line 48-59.
3. In Introduction, added some descriptions on the advances of other fruit juice applied for sparkling wine production.
4. What is “acetic acid sugars” mean in Line 108?
5. Line 123 should be 2.6, line 126, should be 2.7
6. No statistical analysis for pH and Alcohol in Tabel 2?
7. In Figure 4, only sensory analysis results of 10% juice wine. Not base wine for comparison?
Author Response
Please see the attachment.
Comments 1: Line 29, Punica granatum should be italic.
Response 1: We agree with this comment. Therefore, we have modified the sentence as follows: " The pomegranate (Punica granatum) fruit has a round shape and is composed of the exocarp (peel), mesocarp, endocarp and grains.".
Comments 2: Add some reference in Line 48-59.
Response 2: We have added references that tried to compare the difference between the Champagne method and the Martinotti method and their influences on the final wine.
Comments 3: In Introduction, added some descriptions on the advances of other fruit juice applied for sparkling wine production.
Response 3: We have added references about non-grape fruit sparkling wines, particularly those made from plum and apple, and sparkling wines with added fruit juice. On lines 49-51 and have been underlined in yellow.
Comments 4: What is “acetic acid sugars” mean in Line 108?
Response 4: thank you for the comment. It was a typo. We have therefore modified the sentence as follows: “Acetic acid, sugars and acetaldehyde concentrations were quantified using an enzymatic auto analyzer (iMagic-M9, R-Biopharm, Milano, Italy).”
Comments 5: Line 123 should be 2.6, line 126, should be 2.7.
Response 5: thank you for the comment. It was a drafting error. We have corrected this error and have underlined the modification in yellow in the manuscript.
Comments 6: No statistical analysis for pH and Alcohol in Tabel 2?
Response 6: We have added the ANOVA results of these two parameters to Table 2. The added letters are underlined in yellow.
Comments 7: In Figure 4, only sensory analysis results of 10% juice wine. Not base wine for comparison?
Response 7: In the sensory analysis part, we have not made a comparison between the 10% juice wine and the base wine. Because the objective of the sensory evolution was not to compare sensory differences between the different wines. The reason for the analysis was to make a sensory description of the new product and quantify the intensity of the sensory attributes used.

Reviewer 2 Report
Comments and Suggestions for Authors
I read the manuscript 'Pomegranate-Added Wine: A Novel Approach to Sparkling Winemaking', investigating the incorporation of pomegranate juice into the fermentation process to produce a novel rosé sparkling wine. The introduction is short but sufficient, describing the pomegranate as valueable fruit and also about changes in the functional beverages market. The research is well planned. An extensive discussion of the obtained results in relation to those available in the literature
I have some minor suggestions for improvement of your manuscript.
Line 35 - duo -> due
Line 49 - "droad appeal" - some typo?
2.3. Conventional oenological parameter - add literature or specify steps and reagents of the methods
Line 108 - Acetic acid sugars and acetaldehyde -> missing comas?
2.5. Polyphenols analysis - add literature or specify each step of the method
Line 174 - this sentence is unnecessary. Information provided earlier.
Table 1 - acetic acid and volatile acidity measurement methods weren't specify before
Line 237 - "using Kjeldahl" -> Kjeldahl method
Line 243 - "as Bradford still experiences" -> Bradform method or Bradford assay. Similar in line 249.
Sensory evaluation - it's rather a suggestion for the future - when conducting such analyses it's always worth adding a question for the panelists - would they buy such a product? It is also valuable to ask on a hedonistic scale how much they like the taste and aroma of product. The evaluation of the aroma bouquet itself is important for winemakers, but less for most consumers themselves.
I really appreciate all the new reports on functional foods. They have huge potential and should be made available to consumers on a wider scale. And this is one of the best-prepared manuscript in this category of science that I have reviewed in the last year.
Reviewer 3 Report
Comments and Suggestions for Authors
The study aiming to create a healthier wine alternative is interesting, but the relevance and necessity of this product should be emphasized more clearly. The comments below have been noted.
Line 19: What is "balanced" referring to? Based on what?
Line 22: Could you please explain how the inclusion of pomegranate juice can be used as a strategy to counteract climate-related acidity?
Line 23: What types of food products can be classified as functional? I would recommend reconsidering whether the term "functional sparkling wine" is appropriate. What specific functions does it provide, and are these functions genuinely beneficial to consumers' health?
Line 29: The Latin name should be written in italics.
In the introduction, I would recommend emphasizing why this specific use of pomegranate juice was chosen and why it was necessary to supplement traditional wine with it. Are there similar publications on this topic? Additionally, the goal and significance of the study should be highlighted more clearly.
The title of section 2.1 refers to wine, but the description provided only juice.
Line 75: Please provide information about the base wine you are using.
I would recommend presenting the wine production process in a schematic form as well.
Line 117: Please check the citation
Section 2.7: Please provide a brief description of the method used.
Section 2.8: Different sample quantities were used for analysis. Could this have influenced the final results?
Section 2.9: Was the Descriptive Quantitative Analysis (DQA) method employed for sensory profiling? Was the study submitted to and approved by an ethics committee for research involving human subjects? Additionally, were the samples presented in a monadic sequence or as complete blocks?
Figure 1: Please include error bars
Section 3.2: Did you include a control sample without any juice for comparison?
How could the alcohol concentration of the base wine have influenced the alcohol concentration of the wine blended with juice?
Table 3: Please verify if the sample names are correctly labeled, and include letters alongside the data values to indicate statistical significance or group comparisons.
Line 289: "Unfortunately, the multitude of factors influencing colour stability," Was a stability test conducted? If so, please present the data more clearly and provide a more detailed commentary on the results.
Lines 294–295: "The quantities of wine obtained in the initial phase were insufficient for a comprehensive organoleptic test. However, an internal laboratory tasting revealed that in the 15%..." Why was the analysis not repeated? Is this test described in the methods section?
Figure 2: Please provide a more detailed explanation of the results.
Author Response
Please see the attachment.
Comments 1: Line 19: What is "balanced" referring to? Based on what?
Response 1: "Balanced" refers to achieving a harmonious and pleasing combination of acidity, sweetness, and carbonation in the 10% pomegranate juice blend of the sparkling rosé wine. It implies that none of these sensory attributes overpowers the others.
Comments 2: Line 22: Could you please explain how the inclusion of pomegranate juice can be used as a strategy to counteract climate-related acidity?
Response 2: The expression here is not appropriate. Our original intent was to convey that adding pomegranate juice can help increase acidity, which is gradually declining due to climate change. However, we acknowledge that this approach cannot replace other acid-enhancing wine products as a strategy to address decreasing acidity. We have revised the final part of the Abstract accordingly.
Comments 3: Line 23: What types of food products can be classified as functional? I would recommend reconsidering whether the term "functional sparkling wine" is appropriate. What specific functions does it provide, and are these functions genuinely beneficial to consumers' health?
Response 3: We agree with this opinion. Based on your suggestion, we have carefully considered and decided to define the pomegranate juice sparkling wine we developed as an innovative fruit wine with high antioxidant properties. Accordingly, we have revised the final sentences of the Abstract and highlighted the updated content in blue.
Comments 4: Line 29: The Latin name should be written in italics.
Response 4: We agree with this comment. We have already made the modification.
Comments 5: In the introduction, I would recommend emphasizing why this specific use of pomegranate juice was chosen and why it was necessary to supplement traditional wine with it. Are there similar publications on this topic? Additionally, the goal and significance of the study should be highlighted more clearly.
Response 5: The significance of the research lies in these three points: 1) Exploring alternative and innovative uses for pomegranate to maximize its potential beyond traditional applications; 2) Leveraging its unique properties, such as high antioxidant content, to enhance value-added products; 3) Integrating pomegranate juice into sparkling wine production as a way to diversify the portfolio of fruit wines and appeal to modern consumer preferences. Based on these three points, we added a few sentences to the last paragraph of the Introduction. The added sentence is underlined in blue.
Comments 6: The title of section 2.1 refers to wine, but the description provided only juice.
Response 6: We have added the origin of the base wine in Section 2.1. Previously, this information was included in Section 2.2, where the method of sparkling wine production is presented. The modification has been underlined in blue for clarity.
Comments 7: Line 75: Please provide information about the base wine you are using.
Response 7: The information on the Glera base wine is shown in Table 1. On lines 77-78, we have added a sentence to indicate the conventional oenological parameters of the Glera variety base wines shown in Table 1. The added sentence is underlined in blue.
Comments 8: I would recommend presenting the wine production process in a schematic form as well.
Response 8: After careful consideration, we decided not to incorporate this suggestion, as the winemaking process has already been described in detail in Section 2.2, Sparkling Wine Production.
Comments 9: Line 117: Please check the citation
Response 9: We agree with this comment. We have corrected this error in the reference citation.
Comments 10: Section 2.7: Please provide a brief description of the method used.
Response 10: We agree with this comment. We have detailed the method used for protein precipitation and quantification. Now we have the description of the precipitation and quantification method as follows: “Pomegranate juice was filtered at 0.45 µm, and soluble proteins were precipitated and quantified following the KDS method [18]. Protein isolation from 1 mL of sample was achieved by precipitating with 10 µL of 10% Sodium Dodecyl Sulphate (SDS) followed by 250 µL of 1 M KCl. The protein pellet was resuspended in 1 mL of distilled water and quantified using the BCA-200 (Pierce, Rockford, IL) protein assay with a bovine serum albumin (BSA, Sigma, Milan, Italy) standard curve.” The modification has been underlined in blue.
Comments 11: Section 2.8: Different sample quantities were used for analysis. Could this have influenced the final results?
Response 11: The volume of the samples was consistent; however, we initially forgot to include the dilution of pomegranate juice in the method. We have now corrected it as follows: "For each sample, three replicates were prepared by adding 3 μl of pomegranate juice mixed with 7 μl of water, or 10 μl of wine sample, to 2 ml of DPPH solution in a plastic cuvette, ensuring consistent sample volume and preventing the antioxidant activity of pomegranate juice from exceeding the upper limit of measurement." The modification has been highlighted in blue for clarity.
Comments 12: Section 2.9: Was the Descriptive Quantitative Analysis (DQA) method employed for sensory profiling? Was the study submitted to and approved by an ethics committee for research involving human subjects? Additionally, were the samples presented in a monadic sequence or as complete blocks?
Response 12: According to Hawless & Heymann, 2010, p 227-253, the method used in present study is very similar to QDA but not identical. We want to maintain the current description of the method for sensory analysis of 10% pomegranate juice wine. The study was not submitted to the approval of the ethical committee, but we provided the editor with a document of exemption from ethical approval.
Comments 13: Figure 1: Please include error bars
Response 13: We agree with this comment. We have added error bars in Figure 1.
Comments 14: Section 3.2: Did you include a control sample without any juice for comparison?
Response 14: In section 3.2 where the results of the second fermentation of the base wine with different percentages of pomegranate juice in bottles are presented, the fermentations carried out do not include the fermentation with only the base wine.
Comments 15: How could the alcohol concentration of the base wine have influenced the alcohol concentration of the wine blended with juice?
Response 15: When we mix the base wine with different amounts of pomegranate juice, the alcohol content of the base wine contributes to the blend. Consequently, if the base wine has a very high alcohol content, the blend will also have a high alcohol content. This can limit the production of COâ‚‚ during the second fermentation, which is necessary to create the characteristic bubbles. It is important to note that Prosecco typically has an alcohol content of around 11% vol. I mention Prosecco here because the winemaking method used in this study follows the process traditionally used for making Prosecco.
Comments 16: Table 3: Please verify if the sample names are correctly labeled, and include letters alongside the data values to indicate statistical significance or group comparisons.
Response 16: We agree with this comment. We have modified Table 3. The added information is underlined in blue.
Comments 17: Line 289: "Unfortunately, the multitude of factors influencing colour stability," Was a stability test conducted? If so, please present the data more clearly and provide a more detailed commentary on the results.
Response 17: In the present study, we have not performed the analyses on colour stability.
Comments 18: Lines 294–295: "The quantities of wine obtained in the initial phase were insufficient for a comprehensive organoleptic test. However, an internal laboratory tasting revealed that in the 15%..." Why was the analysis not repeated? Is this test described in the methods section?
Response 18: First, in the Materials and Methods section for Sensory Analysis, we have added details about the sensory evaluation of wines fermented in bottles. These additions are underlined in blue. Second, regarding the limited sample size for sensory analysis: fermentation in the bottle was carried out in 375 ml bottles and conducted in duplicate. Each time a sample was taken, some wine was lost due to pressure release when opening the bottle and during filtration prior to analysis, leaving even less for evaluation. Consequently, this part of the study was conducted with a smaller panel of evaluators using a quicker and more flexible sensory analysis method.
Comments 19: Figure 2: Please provide a more detailed explanation of the results.
Response 19: We agree with this comment. We have added further explanations for the colour results. The additions have been underlined in blue.

Round 2
Reviewer 1 Report
Comments and Suggestions for Authors
can be accept